# Neural Stem Cells/Neuronal Precursor Cells and Postmitotic Neuroblasts in Constitutive Neurogenesis and After ,Traumatic Injury to the Mesencephalic Tegmentum of Juvenile Chum Salmon, *Oncorhynchus keta*

**DOI:** 10.3390/brainsci10020065

**Published:** 2020-01-25

**Authors:** Evgeniya V. Pushchina, Ilya A. Kapustyanov, Anatoly A. Varaksin

**Affiliations:** 1Zhirmunsky National Scientific Center of Marine Biology, Far East Branch, Russian Academy of Sciences, Vladivostok 690041, Russia; ilyaak9772@gmail.com (I.A.K.); anvaraksin@mail.ru (A.A.V.); 2Bogomoletz Institute of Physiology, National Academy of Sciences of Ukraine, Kyiv 01024, Ukraine

**Keywords:** traumatic brain injury, neuronal stem cells, cell’s proliferation, neuronal precursor cells, postmitotic neuroblasts, mesencephalic tegmentum, chum salmon, nestin, vimentin, doublecortin

## Abstract

The proliferation of neural stem cells (NSCs)/neuronal precursor cells (NPCs) and the occurrence of postmitotic neuroblasts in the mesencephalic tegmentum of intact juvenile chum salmon, *Oncorhynchus keta*, and at 3 days after a tegmental injury, were studied by immunohistochemical labeling. BrdU+ constitutive progenitor cells located both in the periventricular matrix zone and in deeper subventricular and parenchymal layers of the brain are revealed in the tegmentum of juvenile chum salmon. As a result of traumatic damage to the tegmentum, the proliferation of resident progenitor cells of the neuroepithelial type increases. Nestin-positive and vimentin-positive NPCs and granules located in the periventricular and subventricular matrix zones, as well as in the parenchymal regions of the tegmentum, are revealed in the mesencephalic tegmentum of juvenile chum salmon, which indicates a high level of constructive metabolism and constitutive neurogenesis. The expression of vimentin and nestin in the extracellular space, as well as additionally in the NSCs and NPCs of the neuroepithelial phenotype, which do not express nestin in the control animals, is enhanced during the traumatic process. As a result of the proliferation of such cells in the post-traumatic period, local Nes+ and Vim+ NPCs clusters are formed and become involved in the reparative response. Along with the primary traumatic lesion, which coincides with the injury zone, additional Nes+ and Vim+ secondary lesions are observed to form in the adjacent subventricular and parenchymal zones of the tegmentum. In the lateral tegmentum, the number of doublecortin-positive cells is higher compared to that in the medial tegmentum, which determines the different intensities and rates of neuronal differentiation in the sensory and motor regions of the tegmentum, respectively. In periventricular regions remote from the injury, the expression of doublecortin in single cells and their groups significantly increases compared to that in the damage zone.

## 1. Introduction

Neurogenesis in adult animals is a variable process that can be enhanced by environmental factors and physical activity [1,2] and is modulated by micro-RNA under the influence of external epigenetic mechanisms [3,4,5]. The factors causing the transition between proliferation and differentiation, as well as between neurogenesis and gliogenesis, are still not fully understood. They include external factors such as cytokines, growth factors, neurotransmitters, and morphogens, as well as internal factors, including transcription factors, epigenetic regulators, and non-coding forms of RNA [4,5]. Unlike mammals, adult fish have intense neurogenesis observed in many areas of the brain [6,7]. Systematic anatomical studies on zebrafish [8,9] and medaka [10] identified at least 16 different neurogenic niches in the brain of these species. The pallial and subpallial neurogenic zones identified in the fish brain are homologous to the subgranular zone of the hippocampus (SGZ) and subventricular zone of the lateral ventriculus (SVZ) in the mammalian brain [8,11]. Analysis of viral tracing showed that the radial glia of adult animals is a self-renewing and pluripotent cell population similar in properties to stem cells [12]. In the optic tectum [13] and the cerebellum [13], neural stem cells (NSCs) are not glial in nature and express neuroepithelial markers.

The production of neurons, which is ongoing throughout the postembryonic period of ontogenesis, has not yet been studied, at least in some limited areas of the brain observed in all species of vertebrates. The ratio of adult NSCs (aNSCs), neuronal precursor cells (NPCs), and postmitotic neuroblasts characterizes the early post-proliferative processes of generation of new neurons and requires detailed characterization using appropriate morphogenetic markers. The ratio of neuro- and gliospecific proteins that mark populations of proliferating progenitor cells in the matrix zones of the telencephalon and cerebellum is not similar between different fish species. For example, glial phenotype cells labeled with vimentin are detected in the zebrafish pallium [14]. In the dorsal matrix area of the zebrafish cerebellum, vimentin-positive radial glia are found [14]. In our studies on juvenile salmon fish, no cells of a similar phenotype were detected in the corresponding brain regions.

The tegmental region in the fish brain remains the least studied compared to other brain regions. Earlier, we considered the ratio of proliferation processes, neuronal and glial differentiation in the mesencephalic tegmentum of juvenile chum salmon [15]. Studies have shown the presence of an extensive proliferating cell population in the periventricular region of the tegmentum, labeled with proliferating cell nuclear antigen PCNA. The presence of extensive proliferative zones in the tegmentum indicates a high intensity of adult neuro- and gliogenesis, which has been confirmed by the appropriate immunohistochemical (IHC) labeling of the neuronal protein HuCD and GFAP [15]. For the further characterization of aNSCs/NPCs and a population of postmitotic neuroblasts located both in the primary periventricular zones of the tegmentum and parenchymal regions, these cell types need to be identified using specific markers. Due to the controversial information about the labeling of precursor cells in the brain of various fish species, in our work, two markers of intermediate filaments—vimentin and nestin—were identified in order to determine the phenotype of progenitor cells in the tegmentum of juvenile chum salmon. For the IHC identification of postmitotic neuroblasts appearing during constitutive neurogenesis, as well as after traumatic injury to the tegmentum, doublecortin labeling was used.

The following objectives were set up in the present study: to characterize the proliferative potential of the cells from the periventricular region and parenchyma using experimental labeling of 5-bromo-2’-deoxyurenedine (BrdU) and assess the variation in proliferative activity of cells after a traumatic injury to the tegmentum; to characterize neuronal precursors in the matrix zones of the brain using IHC labeling of vimentin and nestin and study the change in the expression of these markers during a traumatic injury of the mesencephalic tegmentum; to characterize the populations of tegmental postmitotic neuroblasts that are generated during constitutive neurogenesis; and to evaluate the dynamics of the post-traumatic expression of doublecortin in neuroblasts of the mesencephalic tegmentum of juvenile chum salmon.

## 2. Material and Methods

### 2.1. Experimental Animals

We used 40 juvenile (one-year-old) individuals of chum salmon, *Oncorhynchus keta*, with a body length of 10–12.5 cm and weight of 30–45 g. The fish were obtained from the Ryazanovka experimental fish hatchery in 2018. They were kept in a tank with aerated fresh water at a temperature of 16–17 °C and fed once a day. The light/dark cycle was 14/10 h. The content of dissolved oxygen in water was 7–10 mg/dm^3^, which corresponds to normal saturation. All experimental manipulations with animals were carried out in accordance with the rules regulated by the Comission on Biomedical Ethics of the Zhirmunsky National Scientific Center of Marine Biology (NSCMB), Far East Branch, Russian Academy of Sciences (FAB RAS) (approval # 2-221019 from Meeting No. 2 of the Commission on biomedical ethics of National Scientific Center of Marine Biology of the Far Eastern Branch of the Russian Academy of Sciences, October 22, 2019). The fish were anesthetized in a solution of tricaine methanesulfonate MS222 (Sigma, USA) for 10–15 min.

After anesthesia, the intracranial cavity of an immobilized animal was perfused with a 4% paraformaldehyde solution prepared in 0.1 M phosphate buffer (pH 7.2) using a syringe. After prefixation, the brain was extracted from the cranial cavity and fixed in a 4% paraformaldehyde solution for 2 h at 4 °C. Then, it was kept in a 30% sucrose solution at 4 °C for two days (with a five-fold change of solution). Serial frontal 50 μm sections of the chum salmon brain were cut on a Cryo-Star HM 560 MV freezing microtome (Carl Zeiss, Oberkochen, Germany), mounted on gelled glass slides, and dried.

### 2.2. Experimental Damage to the Midbrain Tegmentum

The experimental damage was made by piercing the skull of a fish with a sterile 27G 0.5-inch needle (BD Bioscience, Cat. No 305109). Between the hemispheres of the optical tectum, in the midline of the brain, a 3 mm deep wound was inflicted in the parasagittal direction. The area of injury covered the tectum and the tegmentum of the midbrain and did not affect other parts of the brain. Immediately after the mechanical damage, the animals were released back into the tank (size: 90 × 60 × 50 cm) with fresh water for their recovery and further monitoring.

### 2.3. Immunohistochemistry

For the identification of cell proliferation, we used anti-5-bromo-2’-deoxyurenedine (BrdU) immunohistochemistry (IHC), and the neuronal precursor cells nestin and vimentin IHC [14]. Doublecortin was used for the IHC labeling of postmitotic neuroblasts in the brain of juvenile chum salmon. To study the expression of doublecortin, vimentin, and nestin in the mesencephalic tegmentum of juvenile chum salmon, immunoperoxidase labeling was applied to frozen free-floating sections of the brain. Assessment of the activity of morphogenetic markers was carried out at 3 days after the traumatic injury to the mesencephalic tegmentum.

To identify vimentin, nestin, and doublecortin, frontal 50 μm brain sections were cut on a Cryo-star HM 560 MV (Carl Zeiss, Oberkochen, Germany) freezing microtome. The brain sections were washed thrice in 0.1 M phosphate-buffered saline. The IHC activity of vimentin, nestin, and doublecortin was determined using standard avidin-biotin peroxidase labeling on freely floating sections. The sections of mesencephalic tegmentum were incubated with monoclonal mouse antibodies against vimentin, nestin, and doublecortin (Abcam, Cambridge, UK) (1:300) at 4 °C for 48 h. To visualize the IHC labeling, a standard kit (Vectastain Elite ABC Kit, Burlingame, CA, USA) was used. The red substrate (VIP Substrate Kit, Vector Labs, Burlingame, CA, USA) in combination with methyl green staining according to Brachet’s technique was applied. The material was dehydrated in accordance with the standard protocol in two shifts of ethanol (96%), passed through xylene, and embedded in Biomount C media for histological sections (Biognost, Zagreb, Croatia). 

To assess the specificity of the immunohistochemical reaction, the negative control method was used. Instead of primary antibodies, the brain sections were incubated with a 1% solution of non-immune horse serum at 4 °C for 48 h and treated as sections with primary antibodies. In all the control experiments, no immunopositive reaction was observed.

### 2.4. Experimental BrdU Labeling 

To study the proliferation in intact animals and at 3 days after the traumatic injury of the mesencephalic tegmentum in young chum salmon, we applied the IHC labeling of BrdU. The fish were anesthetized in a cuvette with 0.1% solution of tricaine methane sulfonate MS-222 for 5 min (St Louis, LO, USA). The traumatic injury was a 1 mm deep wound inflicted in the parasagittal direction of the tegmentum. In the control group of animals and after the tegmentum injury (*n* = 5 for each group), an intraperitoneal injection of 10 mg/mL BrdU solution (Sigma-Aldrich, St Louis, LO, USA) at a dose of 20 μL/g body weight was administered to animals simultaneously with brain damage. After a 3 day period, the fish brain was extracted for further analysis. Animals of the control group (*n* = 5) received only BrdU injection. Immediately after the traumatic injury, the animals were released back into the tank with fresh water for their recovery and further monitoring.

For further IHC studies, the fish were anesthetized in a cuvette with 0.1% solution of tricaine methane sulfonate MS-222 for 5 min, the skull was opened, and the brain was extracted. Then, the whole brain was embedded into paraffin according to the generally accepted technique; serial transverse 7 μm sections of the brain were cut and mounted on gelled glass slides. After that, the sections were deparaffinized according to the standard histological protocol. At the final stage, they were washed in distilled water for 3 min. Sections were further processed according to the protocol for IHC labeling of BrdU [16]. To untwist the double-stranded structure of DNA, acid hydrolysis was performed (www.thermofisher.com). The brain sections were incubated in 1 M HCl for 10 min on ice, then incubated in 1 M HCl for 10 min at room temperature, and then for 20 min at 37 °C. Immediately after the incubation with acids, the sections were neutralized in 0.1 M borate buffer for 10 min at room temperature and washed three times in PBS phosphate buffer (pH 7.4), 0.1% Triton X-100, three times, with 5 min per wash. A 1% hydrogen peroxide solution on 0.1 M phosphate buffer was applied to the sections (pH 7.2). The sections were then incubated for 20 min at room temperature and washed thrice in 0.1 M phosphate buffer for 5 min per wash. Subsequently, the sections were incubated with the anti-bromodeoxyuridine/BrdU monoclonal mouse antibody (1:200; clone SPM166; Novus Biologicals, Littleton, MA, USA) at room temperature for 30 min and then washed in three shifts of 0.1 M phosphate buffer for 5 min per shift. To visualize IHC labeling, a standard Vectastain Elite ABC kit (Vector Laboratories, Burlingame, CA, USA) was used according to the manufacturer’s instructions. The red substrate (VIP Substrate Kit, Vector Labs, Burlingame, CA, USA) was used for the visualization of the IHC reaction. The sections were dehydrated according to a standard procedure and enclosed under coverslips in the Biomount C media for histological sections (Biognost, Zagreb, Croatia).

### 2.5. Microscopy

A motorized inverted microscope Axiovert 200 M with an ApoTome module and digital cameras AxioCam MRM and AxioCam HRC (Carl Zeiss, FRG, Oberkochen, Germany) was used to visualize the proliferation, NPCs, and migration of neuroblasts, as well as to conduct morphological and morphometric analysis. Micrographs of the mounts were obtained, and an analysis of the material was carried out using the AxioVision program. The morphometric analysis of the parameters of cell bodies (measurement of the greater and lesser diameters of the soma of cells) was performed using the software supplied with the Axiovert 200 M microscope (Oberkochen, Germany). The measurements of the cell number per field were performed at magnifications of objective 20× and ocular 10× in several randomly selected fields of view for each study area. Previously developed classifications for mesencephalic tegmentum cells of chum salmon [15], along with size characteristics, were used to classify and typify the cells formed during the period of constitutive and reparative processes in the proliferative zones and definitive centers of the mesencephalic tegmentum. Microphotographs of the mounts were obtained using an Axiovert 200 digital camera (Oberkochen, Germany). The material was processed using the Axioimager program and the Corel Photo-Paint 15 graphics editor.

### 2.6. Statistical Analysis

Quantitative processing of the material was performed using the Microsoft Excel 2010 and Statistica 12 software packages STATA statistical software (StataCorp. 2012. Stata Statistical Software: Release 12. College Station, TX: StataCorp LP, USA). The distribution density and dimensional characteristics of cells were assessed using methods of variation statistics. All data were expressed as mean value ± standard deviation (mean ± SD) and analyzed with the SPSS software (version 16.0; SPSS Inc., Chicago, IL, USA). All variables measured in groups were compared using Student’s *t*-test or one-way analysis of variance (ANOVA, Chicago, IL, USA), followed by Newman–Keuls post-hoc analysis. Values at *p* ≤ 0.05 were considered statistically significant.

## 3. Results

### 3.1. Experimental Labeling of BrdU in the Intact Tegmentum of Juvenile Chum Salmon and after Traumatic Injury

At 3 days after the experimental administration of BrdU in the tegmentum of chum salmon, the number of labeled nuclei and cells was estimated. After the immunoperoxidase labeling of BrdU+, the elements were stained brown, while immunonegative cells and mesencephalo-cerebellar tracts were not stained (Figure 1A,B). In the control, BrdU+ cells and, in some cases, BrdU+ nuclei (Figure 1A) were detected in the periventricular zone (PVZ), subventricular zone (SVZ), and parenchymal layers (PZ) of the mesencephalic tegmentum. The morphometric parameters of BrdU+ elements are shown in Table 1. According to the classification of Traniello and co-authors [17], we took into account several types of BrdU-labeled elements: cell nuclei up to 3.5 μm in size and BrdU+ cells of various types with sizes varying from 4.4 to 9 μm (Table 1). BrdU-labeled cells and nuclei were detected in PVZ, SVZ, as well as at different levels of PZ (Figure 1A,B). In the control animals and those after traumatic injury, BrdU+ nuclei had similar morphological characteristics; the BrdU+ cells had a rounded or oval shape (Table 1; Figure 1A,B).

In control animals, BrdU+ nuclei and small rounded cells of 3–4.5 µm in size were observed quite frequently in PVZ (Figure 1A). Sometimes, larger cells that formed small groups were detected (Figure 1A (inset)). In the SVZ and deeper parenchymal layers of the tegmentum, small BrdU+ cells also dominated; their distribution was uniform (Figure 1A).

After the traumatic injury to the tegmentum of juvenile chum salmon, we studied similar zones. The appearance of additional cell types and conglomerates of labeled cells was detected in the PVZ, SVZ, and deep PZ of the tegmentum (Figure 1B). In PVZ, single BrdU+ cells (Figure 1C) and small conglomerates of cells (Figure 1B) were found. In the PZ, the number of BrdU+ cells outside the injury zone increased slightly compared to the control. Along with intensely labeled cells and nuclei, moderately labeled cells were identified in the PZ (Figure 1D). In the SVZ, the number of intensely labeled small and larger oval cells increased; intensely labeled cells were identified as a part of small conglomerates (Figure 1E). A quantitative data analysis showed an increase in the number of BrdU+ cells in the tegmentum PVZ after traumatic injury compared with the control level (Figure 1F).

Along with the areas adjacent to the injury zone, the latter was examined directly (Figure 2A). The characteristics of the cellular composition of BrdU+ and BrdU– in the injury zone are shown in Table 1. In this area, cell types and labeled nuclei that were morphologically heterogeneous were identified (Figure 2B; Table 1). The BrdU labeling intensity also varied from moderate to high (Figure 2B) in the dorsal part of the injury zone. Along with BrdU+, a dense accumulation of BrdU– cells was revealed in the areas adjacent to the injury zone. In the PVZ and SVZ, an increased number of BrdU+ cells was also observed.

In the deeper layers of the PZ, the composition of BrdU+ cells in the injury zone remained heterogeneous (Figure 2B); however, the number of BrdU+ cells in the ventral direction decreased compared to the dorsal layers (Figure 2C). The number of BrdU + cells adjacent to the injury zone of the parenchyma (PZ) did not significantly increase; both single and paired marked cells dominated (Figure 2D). The quantitative data analysis showed an increase in the number of BrdU + cells in the area of injury compared with the control level (Figure 2E, *p* ≤ 0.05). The ratio of BrdU+/BrdU– cells in the PVZ after the injury and in the injury zone is shown in Figure 2E.

In the PZ adjacent to the area of injury, the number of BrdU+ cells did not noticeably increase, both single and paired marked cells dominated (Figure 2D). The quantitative data analysis showed a significant increase in the number of BrdU+ cells in the area of injury compared with the control level (Figure 1F, *p* ≤ 0.05).

### 3.2. Labeling of Neuronal Precursors with Nestin in the Mesencephalic Tegmentum of Juvenile Chum Salmon in Normal Conditions and after Traumatic Injury

The data analysis of the nestin-immunolabeled section of chum salmon tegmentum allows us to identify the small-sized neuronal progenitor cells’ (NPCs) round or oval shape in the territory of the PVZ, SVZ, and PZ (Figure 3A–C). Another type of Nes+ elements was represented by granules of various sizes and shapes, which were detected both inside the cells and in the extracellular space. The morphological parameters of nestin-positive and nestin-negative elements are shown in Table 2.

In the lateral tegmentum in control animals, Nes+ NPCs, as well as granules, were detected in all zones (Figure 3A). In the PVZ, Nes+ granules were identified inside neuroepithelial cells. In the basal part of the PVZ and SVZ, single intensely labeled NCPs were found (Figure 3A, inset). Single Nes+ NPCs were also found in the deeper parenchymal layers of the tegmentum; however, in the PZ, nestin expression dominated the intercellular space in the form of diffuse deposits associated with cell conglomerates or in white matter (Figure 3A).

In the medial tegmentum, the distribution pattern of Nes+ granules and NCPs in the basal part of the PVZ and SVZ was slightly different (Figure 3B). Nes+ granules were more densely located in the basal layer of the PVZ, and immunopositive NPCs were less common than in the lateral tegmentum. In the SVZ, single labeled NPCs prevailed (Figure 3B). In the parenchymal part of the medial tegmentum, clusters of Nes– cells, as well as zones of denser deposition of nestin in the extracellular space, were clearly identified (Figure 3B). The presence of immunonegative accumulations of cells indicates the processes of constitutive neurogenesis in the mesencephalon of juvenile chum salmon. The identification of areas with deposition of nestin in the intercellular space in the parenchymal zone of the medial tegmentum indicates a high intensity of plastic processes in the mesencephalic tegmentum of intact juvenile animals. An increased density of extracellular deposition of nestin-positive granules was also detected in the territory of the dorsomedial tegmentum nuclei (DMTN), in the area containing large immunonegative motor neurons (Figure 3C). In the PZ in this territory, the largest number of immunopositive NPCs was found, significantly exceeding the content of these cells in the PVZ and SVZ (Figure 3C). The spatial distribution of Nes+ NPCs and granules is associated with the presence of the two largest regions of their localization. In the first case, Nes+ cells and granules are detected in the basal part of the PVZ, among the multilayered neuroepithelium that forms the structure of the PVZ. In the second case, the largest accumulation of Nes+ NPCs and granules is associated with undifferentiated immuno-negative cell masses of the SVZ of the lateral tegmentum (Figure 3D) and large definitive cells of the medial tegmentum (Figure 3E). The quantitative data analysis indicated, however, that the largest number of NPCs was revealed in the tegmental PZ compared with the PVZ and SVZ (Figure 3F). The results of the one-way analysis of variance showed significant intergroup differences in the quantitative content of Nes+ NPCs between the PVZ and PZ, as well as the SVZ and PZ (Figure 3F). Thus, the presence of Nes+ NPCs in various regions of the mesencephalic tegmentum, both in the primary periventricular region and in the deep parenchymal layers, as well as the presence of nestin expression in the intercellular space, indicates intense processes of constitutive neurogenesis and high neuronal plasticity in this region of the brain of young chum salmon.

Significant structural changes and pathomorphological changes in the area of the injury were observed on day 3 after the traumatic injury to the tegmentum (Figure 4A). They were manifested both as a local increase in the total number of Nes+ NPCs in the area of injury, and as the appearance of additional Nes+ cell types absent in intact animals (Table 2). In particular, two additional cell types larger than constitutive NCPs appeared. An increase in the intensity of nestin labeling in the intercellular space compared with that of intact animals was evident (Figure 4A). Another structural change is the appearance of additional clusters containing Nes+ cells and extracellular deposition of nestin-positive granules in the SVZ adjacent to the injury zone (Figure 4B).

Such regions contained dense clusters of small NPCs and larger Nes+ cells of the third and fourth types (Table 2), alternating with zones of intense deposition of nestin-positive granules in the intercellular space (Figure 4B). At a certain distance from such areas, they were identified as nestin-immunonegative (Nes–) reactive clusters of cells, containing, however, intracellular deposits of nestin in the form of granules (Figure 4B) and local clusters of large Nes+ cells of the fourth type. A significant increase in the number of Nes+ cells of all types, substantially exceeding their number in the PZ of control animals, was observed immediately in the injury area (Figure 4F).

In the injury zone, we also revealed single cells and their local clusters, which formed cell arrays of considerable length (Figure 4C). In the central part of the injury, local clusters and single Nes+ NPCs were located along the guiding fibers of the radial glia, oriented inside the injury zone and forming the trajectories of the cells to their places of final localization. Similarly, patterns of NPCs localization were found ventrally in the peripheral part of the injury, the guides for which were presumably the fibers of Nes-immunonegative radial glia (Figure 4C). Local immunonegative cell clusters containing intracellular Nes+ granules and individual small clusters of Nes+ cells were found on the periphery of the injury zone, in its dorsal part (Figure 4C). In the SVZ, a significant increase in the number of Nes+ NPCs (*p* ≤ 0.05) compared with the control was revealed (Figure 4F). In the tegmentum areas not associated with injury, in the SVZ, single type 4 Nes+ cells, as well as reactive groups of negative cells containing Nes+ granules, were found (Figure 4D). In the PVZ of such areas, there was an increased density of distribution of single Nes+ NPCs and their clusters, as well as abundant intracellular and extracellular Nes+ granules (Figure 4D). In the PZ adjacent to the injury zone, we observed a similar pattern of distribution of Nes+ NPCs and granules (Figure 4E), which formed areas of increased distribution density.

Thus, on day 3 post-injury, a series of pathomorphological changes occurred, which was associated with the increase in the expression of nestin in NPCs. The appearance of additional types of Nes+ cells participating in the traumatic process induced the proliferation of progenitor cells both inside the traumatic focus and in additional nestin-producing regions in the SVZ and remote parenchymal zones of the tegmentum.

### 3.3. Vimentin Labeling of Progenitor Cells in the Mesencephalic Tegmentum in Intact Juvenile Chum Salmon and after Traumatic Injury

In the control animals, the IHC expression of vimentin—an intermediate filament protein used as a marker of astrocytic glia [14]—was detected in the medial and lateral regions of the tegmentum (Figure 5A,B). After labeling with vimentin (Vim), we identified three types of cells, as well as subcellular vimentin-positive granules whose morphological parameters are shown in Table 3. Small, intensely labeled cells were identified in the PVZ, SVZ, and PZ of the medial (Figure 5A) and lateral (Figure 5B) parts of the tegmental divisions. Larger Vim+ cells were detected in the PVZ/SVZ of the lateral tegmentum (Figure 5B). Intensely and moderately Vim+ labeled granules, localized both inside the cell and in the intercellular space, were detected everywhere, both in the form of separate discrete granules and in the composition of extended regions with diffuse localization (Figure 5A–C). The concentration of Vim+ cells in the PVZ and SVZ of the control animals was rather low, and in the PZ, a relatively high concentration of Vim+ cells was revealed. The quantitative data analysis showed significant intergroup differences in the concentration of Vim+ cells and granules in the PVZ, SVZ, and PZ of the tegmentum in control groups (Figure 5F). Constitutive clusters of Vim– cells (Figure 5C), forming local neurogenic niches characterizing the high intensity of constitutive neurogenesis in the tegmentum parenchyma (Figure 5A,B), were comparatively frequently determined in parenchymal regions. Similar cell groups were also detected in the SVZ; in the PVZ, neuroepithelial type Vim+ cells were mainly identified, among which there were individual Vim+ cells and granules (Figure 5D). Dense areas of extracellular deposition of vimentin, as well as an increased density of distribution of Vim+ granules (Figure 5E), were found in the DMTN, in the PVZ and SVZ.

On day 3 post-injury, the number of Vim+ cells and granules increased in the medial and lateral parts of the tegmentum (Figure 6A,B). Despite the increase in the number of Vim+ cells in the PVZ and SVZ, only cells of types two and three were detected, while larger type-four cells were not found (Table 3). The increase in the Vim-immunopositivity of the tegmentum after injury was mainly due to the increased expression of vimentin in the intercellular space (Figure 6A,B). Patterns of Vim localization with increased number of granules that formed intracellular and extracellular deposits with an increased distribution density in the PVZ and SVZ were most typical in the post-injury period (Figure 6A,B). In the PVZ adjacent to the injury area, patterns of increased Vim– cells distribution density showed the characteristic growth. Single Vim+ granules, as well as small conglomerates of Vim+ cells, were visualized as a part of such areas (Figure 6C).

In the area of injury, we identified multiple foci of localization of Vim+ cells and their conglomerates surrounded by areas with increased extracellular expression of vimentin (Figure 6D). We found at least three to four foci of Vim-immunopositivity located in the territory of the PVZ and SVZ, as well as spreading into the deep parenchymal layers (Figure 6D). Along with such foci, Vim+ cells were clearly identified in the basal part of the PVZ and SVZ adjacent to the injury area (Figure 6D). In deep parenchymal layers, outside the foci, we observed local accumulations of Vim+ cells with no increased extracellular expression of vimentin around them (Figure 6D). Such clusters looked like local clusters of cells surrounded by an immunonegative space (Figure 6D). The foci of Vim positivity found in the area of injury contained immunopositive cells and their clusters, which were surrounded by an extracellular space with increased vimentin expression (Figure 6E). They looked like dark, immunopositive, extended areas that occupied a significant portion in the near-traumatic space and, obviously, were characterized by a high intensity of regenerative processes. The extracellular deposition of vimentin in such areas was diffuse or granular in nature; in some cases, thin fibrillar structures occurred.

A large additional focus of Vim positivity, with a high level of vimentin expression, was detected in the ventral part of the SVZ (Figure 6D,F). The morphological structure of this lesion was similar to that of other near-injury lesions (Figure 6D–F). However, the plane of localization of cells in this focus was perpendicular to the direction of the wound (Figure 6D) and looked quite specific. Most Vim+ cells were oriented horizontally, and reactive Vim clusters of cells were located immediately below them (Figure 6F).

The lesion closest to the injury area contained several local clusters of Vim+ cells surrounded by single Vim– cells (Figure 6G). In this cluster, the distribution density of labeled cells and the expression of extracellular vimentin were reduced compared to other foci (Figure 6E–G). The quantitative data analysis showed a significant increase (*p* ≤ 0.05) in the proportion of Vim+ cells in the injury area (Figure 6H).

### 3.4. Expression of Doublecortin in Postmitotic Neuroblasts of the Tegmentum in Intact Juvenile Chum Salmon and after a Traumatic Injury

The expression of doublecortin (DC) associated with microtubules of protein was detected in a heterogeneous population of tegmentum postmitotic neuroblasts represented by three morphological cell types, whose parameters are shown in Table 4. Along with DC+ cells, we also found subcellular DC+ granules (Figure 7A). The maximum distribution density of the granules was revealed in the medial part of the tegmentum; the minimum was found in the dorsal part (Figure 7A). The density of DC+ cells in the PVZ of the dorsal tegmentum was relatively low (Figure 7B). The number of DC+ cells in the PVZ and SVZ, as well as DC+ granules in the SVZ, increased slightly in the lateral tegmentum region (Figure 7C). In the DMTN region, small clusters of DC+ cells were identified in the PVZ (Figure 7D); they were more numerous than in lateral tegmentum. In this area, single labeled cells and more abundant DC+ granules in the SVZ (Figure 7D) were also found. In the parenchymal layers, the distribution density of DC+ cells and granules, as well as their proportion in the total number of cells, was at maximum compared with those in the PVZ and SVZ (Figure 7F). In the PZ, DC+ granules were especially frequently surrounded by clusters of large motor neurons in the reticular formation and adjacent clusters of DC– cells forming constitutive neurogenic niches (Figure 7E). The ANOVA revealed significant intergroup differences in the distribution of DC+ cells in the PVZ and SVZ, as well as in the PZ (Figure 7F).

After the traumatic injury, the number of DC+ neuroblasts increased significantly compared to their number in the control animals. In different zones of the tegmentum, clusters of postmitotic DC+ cells and their groups were detected (Figure 8A). Such clusters, as a rule, were localized in the PVZ and/or SVZ (Figure 8A). In the deeper parenchymal regions of the tegmentum, the number of DC+ cells also increased significantly (Figure 8B). The morphometric parameters of DC+ cells and granules that appeared on day 3 post-injury are shown in Table 4. The number of DC+ cells in PVZ after damage significantly increased (*p* ≤ 0.05) compared with their number in the control animals (Figure 8G).

In the area of traumatic injury, the distribution patterns of DC+ and DC– cells differed significantly from those in the control animals (Figure 8C). In the injury area, the hypertrophy of MG-stained DC– cells was clearly visualized in the areas of PVZ and SVZ (Figure 8C,D). Obviously, in these areas in the post-injury period, the reactivation of resident “silent” NSCs was observed, followed by the implementation of multiple proliferative cycles of progenitor cells. The number of postmitotic neuroblasts labeled with DC was relatively small (Figure 8D), which indicates the predominance of proliferation processes over early cell differentiation in this area. This is also evidenced by the presence of large DC– reactive clusters of cells located ventrally within the injury region (Figure 8D). The number of DC+ neuroblasts in the injury area was not high; however, their number increased significantly as they deepened into the PZ and the adjacent near-injury zones (Figure 8C). The distribution of DC+ granules in these areas was also increased (Figure 8D), which indicates the intensification of the processes of early neuronal differentiation in the tegmentum PZ. In the ventral part of the traumatic zone, the content of DC+ granules was also high; they were located among DC– cells, forming patterns of extracellular expression (Figure 8E). In the areas of PVZ adjacent to the injury zone, areas of hypertrophied neuroepithelium with increased distribution density of DC+ neuroblasts were identified (Figure 8F). Quantitative analysis data indicate a significant increase in the number of DC+ neuroblasts in the parenchymal regions of the injury zone in the tegmentum (*p* ≤ 0.05) compared to control values (Figure 8G). Thus, on day 3 post-injury, the number of DC+ postmitotic neuroblasts rises in the PVZ, as well as in the parenchymal layers of the tegmentum surrounding the injury area. Large clusters of DC+ neuroblasts, which were absent in the intact animals appeared in the PVZ. In the near-injury areas, the induction of doublecortin expression in the form of an increased content of immunopositive granules was detected, indicating the intensification of cell migration in the post-injury period. Along with a local increase in the number of DC+ neuroblasts in the tegmentum of juvenile chum, after the injury, we also detected hypertrophic zones containing DC– neuroepithelium in which zones of increased content of DC+ neuroblasts were found. We consider such sites as additional zones of post-injury cell proliferation and initial neuronal differentiation of cells formed in the post-injury period.

The ratio of immuno-labeling of vimentin, nestin, and doublecortin in the PVZ, SVZ, and PZ in the tegmentum of intact animals is shown in Figure 9A. After the traumatic injury, the Nes/Vim/DC ratio changed significantly in the SVZ and PZ (Figure 9B). The proportion of Nes+ cells increased and that of DC decreased in the SVZ (Figure 9B). The results of the ANOVA showed that, after the traumatic injury, the labeling ratios of Nes/BrdU/DC (Figure 9C) and Vim/BrdU/DC (Figure 9D) in the PVZ, SVZ, and PZ have significant intergroup differences.

## 4. Discussion

### 4.1. Investigation of the Proliferative Potential of the Mesencephalic Tegmentum of Juvenile Chum with Experimental BrdU Labeling

To study the cellular and molecular mechanisms underlying the regenerative capacity of fish, various models of brain damage in adult animals have been developed [8,9,18,19,20]. After BrdU labeling in the intact mesencephalic tegmentum of juvenile chum salmon, as well as within 3 days after the traumatic injury, cells in the S-phase of cell cycle were detected. As a result of the study, a heterogeneous population of BrdU+ cells and nuclei was revealed in the tegmentum of juvenile chum salmon. The pattern of distribution of BrdU+ cells in intact animals indicates a high intensity of the constitutive neurogenesis processes that occur not only in the matrix proliferative zone of the midbrain located in the PVZ, but also in the deep parenchymal layers of the tegmentum. The presence of proliferating cells in the tegmentum parenchyma of chum salmon confirms the data obtained earlier on the *Apertonotus* [21], according to which single proliferating cells were also found in the cerebellar parenchyma of this fish species. The data obtained on juvenile chum salmon can be interpreted taking into account the specifics of the ontogenetic development of juvenile salmon; in particular, fetalization, which is characterized by a slowdown in the ontogenesis of certain organs or their systems, as a result of which the adult organism retains the embryonic state of the corresponding characters [22]. Fetalization processes coincide in time with the stage of active growth, in which the processes of morphogenesis are most clearly and fully expressed. The group of salmon fish is a phylogenetically ancient branch of vertebrates, which are characterized by a high concentration of undifferentiated elements not only in the matrix zones, but also in the brain parenchyma. Our discovery of BrdU+ cells in the PVZ, SVZ, and PZ of the mesencephalic tegmentum of the intact juvenile chum confirms the previously advanced hypothesis and testifies to the high proliferative potential of NSCs in the mesencephalic tegmentum of the growing juvenile chum salmon.

Our earlier study of PCNA distribution [15] is confirmed by the current results of BrdU labeling in the mesencephalic tegmentum of juvenile chum salmon. However, the number of BrdU+ cells detected in the brain of intact juvenile chum salmon after immunolabeling is much smaller than in case of PCNA labeling. This is explained by both the methodological aspects of the study (PCNA-labeling was performed on thicker frozen sections of the brain, and BrdU labeling was performed on thin paraffin sections) and the features of the detected stages of the cell cycle. According to Wullimann and Puelles, PCNA labels an additional DNA polymerase δ that remains in the cell for 24 h after the completion of mitosis [23], but the level of PCNA activity decreases by 30% [24]. This makes it possible to label cells that are both in the state of proliferation and have recently emerged from mitosis and are in a state of migration. During BrdU labeling, cells in the S-phase of the cell cycle, lasting for 1 to 6 h, are revealed [23]. Thus, the number of cells identified by BrdU labeling in the intact tegmentum of juvenile chum salmon is significantly lower than in the case of PCNA labeling [15].

The proliferative activity of the mesencephalic matrix zones was studied by PCNA labeling in *Carassius carassius* [25]. It was found that the patterns of mitotically active brain cells in this species form morphogenetic fields, referred to as matrix zones. The presence of such fields is characteristic only of bony fish; they were not detected in amphibians and reptiles [25]. Other proliferative zones of the brain were identified in other fish species: three-spined stickleback [7], *Apteronotus* [26,27], *Danio rerio* [28], and *Oncorhynchus masou* [22].

After a traumatic injury, the patterns of BrdU labeling of cells both in the matrix PVZ and in the deeper subventricular and parenchymal layers of tegmentum chum salmon were changed. Thus, small clusters of BrdU-labeled cells were observed to appear in the PVZ and deep layers of the tegmentum; there was also an increase in the number of BrdU+ cells in the PVZ and the appearance of additional, larger types of BrdU+ cells not detected in intact animals. The appearance of clusters of BrdU+ cells indicates the synchronization of proliferative activity in cells, which are most likely the descendants of aNSCs of the neuroepithelial type. In studies on zebrafish, it was found that, as a result of injury, neuroepithelial progenitor cells begin to divide synchronously, forming a pool of newly formed progenitor cells involved in the repair process, some of which have the ability to migrate over long distances [29]. Studies on juvenile chum salmon showed that BrdU+ clusters of cells appear as a result of injury not only in the periventricular zone, but also in the deep regions of the brain, which indicates a high reparative potential of the tegmentum.

The most pronounced changes in the structural organization and the start of additional proliferative activity of cells, whose BrdU labeling was not detected in control animals, are observed in the injury area (Table 1). The different intensities of the BrdU labeling in the injury area indicate that the proliferating cells are not synchronized in time. The morphological heterogeneity of BrdU+ cells in the injury area, in our opinion, may indicate different sources of origin of such cells. Studies on *Danio rerio* have shown that the reactivation of specific genetic programs in resident aNSCs, leading to the activation of proliferative processes, appears after damage [29]. We believe that, in the area of tegmental injury in chum salmon, processes of activation of aNSCs also occur. As a result, there is a local increase in the proliferative activity of aNSCs of different types and possibly of different origins. BrdU labeling in larger cells of injured tegmentum of chum salmon, which is absent in areas of constitutive neurogenesis, may indicate the reprogramming and transdifferentiation of such cells as a result of trauma. This hypothesis is supported by recent studies on zebrafish, according to which mature neurons can transdifferentiate and form a pool of proliferating cells after tectum damage [30]. Such direct conversion or transdifferentiation never appears in the adult brain, which can mean that either aNSCs in the zebrafish brain have a similar unique ability [31], or that this direct transformation has not yet been found in the adult brain of other species due to the lack of a suitable methodology, such as, in particular, in vivo imaging.

### 4.2. Investigation of Neuronal Progenitor Cells in the Tegmentum of Intact Juvenile Chum Salmon and after Traumatic Injury

Nestin is an intermediate type VI fibrillar protein, which is mainly found in postmitotic cells in the central nervous system of vertebrates [32]. Nestin is often used for the IHC labeling of progenitor cells and radial glia [18], which are characterized by the capability of self-renewal and multipotency. In fishes, NPCs are characterized by a high potential for cell renewal after injury [33]. The mechanism of nestin activity in cells at the molecular level is still unclear, despite some functional properties of this protein having been established [32]. Thus, the distribution of nestin-positive cells and patterns of nestin localization in the fish brain can provide an understanding of the molecular cellular mechanisms underlying the high plasticity of the fish brain.

The function of nestin that ensures the processes of constitutive neurogenesis in the brain and the additional expression of nestin observed in the fish brain after injury can be a key to understanding the processes of cell regeneration. Nestin-positive cells in the process of cell differentiation give rise to both neurons and glial cells [32,34]. Unlike mammals, salmonids have numerous neurogenic regions containing a large number of precursors and aNSCs [22]. Proliferative zones in fish are located in the periventricular zones of the telencephalon and diencephalon, as well as at the borders of the midbrain and hindbrain [35]. 

Constitutive neurogenesis is involved in continuous brain growth or renewal of neurons in different functional conditions in animals [36,37]. NSCs of adult animals, including latent NSCs in non-neurogenic regions of the brain, can also be mobilized for regeneration in response to damage of central nervous system [36]. In particular, many non-mammalian vertebrates are able to effectively repair damaged brain by mobilizing NSCs and progenitor cells [36]. Specific differences in regenerative capacity and an increase in neurogenic regions in vertebrates seem to correlate with phylogenetic relationships [37]. These differences may be partly due to the differing potential of aNSCs. In fishes, the regeneration of motor neurons and dopaminergic neurons can be induced in the midbrain of a newborn animal and, accordingly, after a specific injury [21]. 

In a study of nestin-immunopositive regions in the mesencephalic tegmentum, juvenile chum salmon were found to have a rather high level of its expression in NPCs, which was revealed in many areas of the lateral, medial, and median parts of the tegmentum. Large numbers of Nes+ precursor cells with a neuroepithelial phenotype and localized in the periventricular and subventricular regions were also present in the PZ of tegmentum, accounting for about 27% of the cells (Figure 3F). Such a significant content of NPCs in the parenchymal part indicates a high level of persistent neurogenesis characterized by the presence of a large number of constitutive neurogenic niches, which are especially abundant in the territory of the medial tegmentum (Figure 3B). In our studies, patterns of extracellular expression of nestin were revealed both in the form of single granules having subcellular dimensional characteristics, as well as rather extended regions located in the tegmental parenchyma and in the territory of DMTN (Figure 3B,C). We associate such Nes+ deposition zones exhibiting processes of intensive constructive metabolism with the high plasticity providing processes of intracellular transport and renewal in definitive neurons. In studies of functional aspects, it was found that nestin affects the intracellular transport of particles such as vesicles, thus, providing functional support during cell proliferation [38,39]. The separation and transfer of intermediate cytoskeletal filaments during mitosis is performed with the involvement of nestin [35,38]. Thus, nestin plays a significant role in the reorganization of the cytoskeleton components, regulating cellular dynamics by polymerization and/or dismantling of intermediate filaments [32]. However, the molecular machinery of nestin is still unclear, despite some functional characteristics of this protein having been established. In this regard, the study of patterns of localization of nestin in the primary proliferative zones of the mesencephalic tegmentum of juvenile chum salmon—of which the main zone is the region of the mesencephalon–hindbrain border—is extremely relevant. A large number of Nes+ progenitor cells that ensure the processes of constitutive growth of chum salmon tegmentum were found in the PVZ and SVZ. The presence of a large number of Nes+ granules in the cells of the basal part of the PVZ provides a high level of constructive metabolism, including NPCs.

The studies of the mammalian brain have shown that nestin is a marker of radial glia [18,39]. However, in the mesencephalic tegmentum of intact juvenile chum salmon, Nes+ radial glia were not found. In contrast to studies on mammals, the results on juvenile chum salmon showed the presence of Nes+ cells not only in the matrix zones, but also outside them; in particular, in the parenchymal layers of the tegmentum, where the concentration of such elements was significantly higher than the number of Nes+ cells in the PVZ. Thus, the PZ of juvenile chum salmon should be considered in terms of the neurogenic potential comparable to that in PVZ, taking into account the fact that nestin is not expressed by mature neurons [40], being replaced by neuron-specific and glio-specific proteins in mature cells [41].

After a traumatic injury to the tegmentum of juvenile chum salmon, the expression of nestin in its cells increased significantly. We consider an increase in the variety of morphological types of cells expressing nestin and a multiple increase in their number in the area of injury and adjacent areas as the main feature of the post-injury process. Along with the injury zone, secondary foci of nestin-expressing cells were detected in the adjacent SVZ and PZ, which indicates the generalization of the reparative response and the formation of several large reactive zones (Figure 4 A). The local clusters of nestin-positive cells, present both in the injury area and in the secondary foci, indicate the proliferation of NPCs accompanied by the occurrence of nestin-positive cell groups with high proliferative/regenerative potential. Such clusters were formed mainly by cells with similar morphological characteristics (cells of three to four types), which allows us to consider them as descendants of NPCs, activated by the traumatic process and absent in control animals. Such cells were larger than NPCs identified in the tegmentum of intact juvenile chum salmon.

Another feature is the appearance of a large number of nestin-negative cell groups, containing, however, nestin-positive granules inside the cells in the SVZ. Deposits of intracellular nestin are associated with the reorganization of the components of cell cytoskeleton and, apparently, reflect the mitotic activity and mobility of these cells. Such dynamic units, along with dense nestin-positive clusters surrounding the injury area, indicate an active post-traumatic reorganization of the cellular microenvironment in this area. We interpret the detection of longitudinal guides located near the injury area and piercing the damage zone as the appearance of post-traumatic nestin-negative/GFAP+ radial glia, whose presence was described earlier [15]. Migrating nestin-positive NPCs are located along the RG fibers. The heterogeneous composition of nestin-positive cells in the injury area and secondary nestin-positive foci in the SVZ and PZ indicate a large number of nestin-positive cells involved in the post-injury process, among which larger cells appear only after the injury. The quantitative data analysis indicates a significant (2.2-fold) increase in the number of nestin-positive cells in the SVZ and a twofold increase in the number of nestin-positive cells in the injury area (Figure 4F), which amounts to approximately 60%. Thus, as a result of the traumatic process, a reparative response arises from nestin-positive NPCs. First, nestin expression is induced in two additional cell types not expressing nestin in intact animals. As a result of the proliferation of such cells in the post-injury period, local nestin-positive NPCs clusters participating in the reparative response are formed. Second, along with the primary nestin-positive traumatic focus, coinciding with the puncture zone, the formation of additional nestin-positive secondary foci is observed in the adjacent SVZ and PZ. Nestin-negative radial glia, directing nestin-positive NPCs to the regeneration zone, are found in the primary traumatic area.

### 4.3. Expression of Vimentin in Intact Juvenile Chum Salmon and a Change in Constructive Metabolism during a Traumatic Injury to the Tegmentum

Vimentin is an intermediate filament protein expressed by astroglial cells [42], which is regarded as a universal marker of astrocytic glia in the brain of vertebrates [14]. Vimentin, like nestin, is a component of the aNSC and NPC cytoskeleton [43]. In immunohistochemical studies of ependymal cells and radial glia, which are present in large quantities in the fish brain, the presence of GFAP [44] and vimentin [45] was proven. Bony fish vimentin sequencing data and amino acid sequence analysis showed a high degree of homology with human protein [45]. A study on juvenile and adult grey mullet showed that, with age, vimentin levels decrease, while the GFAP expression, in contrast, increases [14]. This is consistent with our data on juvenile chum salmon, according to which, in the juvenile chum salmon tegmentum, the GFAP expression is more pronounced [15] compared with vimentin expression (shown by the results of this work). Comparative studies on vertebrates have shown that, during the development of the central nervous system, vimentin is replaced by GFAP in reptiles [46], birds, and mammals [47]. 

The IHC labeling of vimentin in the tegmentum of juvenile chum salmon revealed three types of cells: small rounded or oval intensely labeled cells similar in morphological characteristics to Nes+ NPCs (Table 2 and Table 3). Two other types of cells (types three and four) are oval-shaped cells with different localizations in the tegmentum; it is possible that type four cells are later developmental stages of type three cells. We believe that, in the chum salmon tegmentum, vimentin reveals NPCs, which is consistent with the results of studies by other authors [43]. In studies on adult zebrafish, GFAP+, BLBP+, S100b+, and Vim+ radial glia cells have been identified near the dorsal matrix zone of the cerebellum, in which granular neurons are generated [48]. 

The maximal part of Vim+ elements in the intact tegmentum of juvenile chum salmon was represented by Vim+ granules of various sizes and distribution densities, with an especially large number of them revealed in the PZ near constitutive Vim– clusters of cells (Figure 5B). Such granules were a component of cell cytoplasm, and were also detected in the extracellular matrix. The diffuse deposits in the form of thin fibrillar filaments, forming zones of various lengths found in all areas of the tegmentum, should be considered another form of extracellular expression of vimentin (Figure 5A–C). In the area of localization of DMTN interneurons, characterized by a high intensity of constructive metabolism, similar zones containing fibrillar vimentin were revealed (Figure 5E). Thus, we believe that a significant amount of Vim+ elements in the mesencephalic tegmentum of intact chum salmon indicates a high intensity of the constructive metabolism processes and constitutive neurogenesis. The quantitative data analysis confirms the presence of a large number of elements in the PZ of the tegmentum, the number of which significantly exceeds that in the PVZ and SVZ (Figure 5F).

After a traumatic injury to the chum salmon tegmentum, we observed a significant increase in the number of Vim+ cells and granules. An increase in the number of Vim+ cells was characteristic of the PVZ and SVZ of the damaged tegmentum. The population of Vim+ cells after the injury was more homogeneous than that in intact animals and represented by cells of types two and three (Table 3). We believe that, as a result of the traumatic process, an additional pool of Vim+ NSC and NPCs was activated in the tegmentum of chum salmon, aiming at eliminating the consequences of the injury. The number of Vim+ granules in the PVZ and SVZ also increased; their distribution density also increased in comparison with the control. The enhanced vimentin expression at the intracellular level was accompanied by labeling of Vim+ granules in the dense reactive cell conglomerates found in the SVZ (Figure 6A–C). In the PVZ, we observed a significant reorganization of the neuroepithelial layer, expressed as the local hypertrophy of areas with neuroepithelial cells, inside which Vim+ granules were also detected.

In the injury area, we detected a significant increase in Vim-expression, both in the composition of numerous labeled cells and their groups, and in the surrounding intercellular space. The results obtained for the juvenile chum salmon tegmentum agree with the data obtained by nestin labeling (present results), as well as with the results of damage to the zebrafish pallium [48]. In previous studies, it was found that traumatic damage is accompanied by the development of acute gliosis, during which astrocytes show increased expression of intermediate filament proteins (GFAP, vimentin, and nestin), and the expression of many genes changes. Damage to the zebrafish pallium enhances the expression of GFAP, vimentin, nestin, and calcium-binding protein S100b in radial glia cells, the processes of which become hypertrophied [48]. Similar structural changes in radial glia cells were revealed by us after GFAP-immunolabeling in the tegmentum of juvenile chum salmon after injury [15]. 

The changes in the expression of GFAP and vimentin in the tegmentum of chum salmon juveniles are similar to gliosis in mammals; however, scar formation in chum salmon juveniles is weak, which is consistent with data on other fish species [18,49]. According to the data of Takeda and his colleagues, GFAP+ processes of radial glia of fish repair damaged axons more than forming any scar [50]. In our studies of chum salmon tegmentum injury, we revealed a significant increase in vimentin expression with the occurrence of large Vim-immunopositive domains, including Vim+ cells and their clusters, surrounded by the zone with an extracellular expression of vimentin. Such large areas were localized both immediately near the injury area and at a larger distance in the parenchymal layers (Figure 6D). A detailed study of the cellular composition and patterns of extracellular immunolocalization of vimentin in these areas showed a multifold increase in cell metabolism. 

Another important conclusion is the multifold increases in the number of Vim+ NPCs after injury. The presence of local clusters of intensely Vim-labeled cells which are similar in size and shape indicates the proliferation of progenitor cells, increasing the total number of cells involved in the reparative process. Thus, the post-injury patterns of vimentin expression are largely reminiscent of nestin expression, which indicates a similar increase in the expression of both intermediate filament proteins in the damaged tegmentum of juvenile chum salmon. As a result of our studies, Vim+ NCPs with neuroepithelial morphology were identified in the injury area. Cells with a radial glia phenotype were not detected.

### 4.4. The Expression of Doublecortin in Neuroblasts of the Mesencephalic Tegmentum Formed during Constitutive Neurogenesis and the Induction of DC Immunopositivity in Post-Injury Neuroblasts of Chum Salmon

Doublecortin is an early neuronal differentiation protein expressed in neuroblasts during development [51], as well as detected in neurogenic regions of the brain in adult animals [52]. Since DC is associated with the components of the cytoskeleton, its expression is also detected during synaptogenesis and during the growth and regeneration of axons. Intracellular doublecortin is localized both in the cytoplasm and in the nuclei of cells [53]. Patterns of extracellular localization of doublecortin, which is a structural component of nervous tissue in mammals and other vertebrates, are known [54]. The most important property of doublecortin is its capability of expression in newborn neurons [55], which makes DC a convenient morphogenetic marker in studies of postembryonic and post-traumatic neurogenesis. Comparative studies investigated patterns of postembryonic expression of doublecortin in the pallial neurogenic zone of zebrafish and *Nothobranchius furzeri*, as well as in the optic tectum in adults of these species [56]. Further studies showed that, in *N. furzeri*, doublecortin expression decreases with age, which is consistent with data obtained on mammals [57]. A study of DC distributions in various fish species, including *Carassius auratus*, *Cyprinus carpio*, and *Salmo gardneri*, showed the local distribution of DC in the migrating cells of the olfactory bulbs [58]. The expression of DC is often associated with the expression of calcium-binding protein calretinin [59]. It is suggested that, in mature neurons, the DC expression provides the neuronal plasticity of the cells that forms the structure of local neuronal networks. It is possible that a decrease in the mitotic activity of cells with age occurs simultaneously with a decrease in DC expression in the brain [60]. 

Doublecortin has been shown to have a similar structure in mammalian and fish brains [52]. Thus, DC+ neuroblasts in the fish brain can be considered as a convenient model for studying the processes of the constitutive migration of postmitotic neuroblasts and post-traumatic changes observed in the brain of various vertebrates.

A study of the localization of doublecortin in the tegmentum of intact juvenile chum salmon showed that this protein is moderately expressed in cells of primary neurogenic zones during constitutive neurogenesis. Compared with other intermediate filament proteins—vimentin and nestin—the number of DC+ cells is slightly reduced in the PVZ of the juvenile chum salmon tegmentum, which probably indicates a relatively low number of postmitotic neuroblasts in the primary proliferative zone. However, in different regions of the tegmentum, the number of DC+ neuroblasts somewhat differs, with the predominance of such cells in the lateral region of the tegmentum. In the lateral tegmentum, there are a large number of undifferentiated cells that form the cell masses of the *torus semicircularis* and the lateral nuclei of *valvula cerebelli*, which exhibit a pronounced production of postmitotic cells. Obviously, postmitotic neuroblasts transform into neurons of the sensory regions of the *torus semicircularis* or form the structure of evolutionarily young and variable *valvula cerebelli*.

In the medial part of the tegmentum, on the contrary, motoneurons of DMTN are concentrated, which coordinate motor activity; they are functionally mature in very young chum salmon after hatching. In these areas, we observed an increased distribution of DC+ granules with both intracellular and extracellular localization. In the PZ of the tegmentum, among the large interneurons of the reticular formation, an increased content of DC+ cells was revealed, which indicates intensive processes of neuronal differentiation in the tegmental region of the chum salmon brain. Thus, the distribution patterns of DC+ accumulations of cells indicate intensive processes of constitutive neurogenesis in the SVZ and PZ of the juvenile chum salmon tegmentum, which agrees with the results of Vim and Nes immunolabeling. In the PVZ, on the contrary, no intense expression of DC was detected in neuroblasts and granules, which indicates the predominance of proliferative processes in this area of the tegmentum.

After the traumatic injury, a significant change in the DC-immunolocalization pattern cells was revealed in the PVZ. We found both local accumulations of DC+ cells and their groups in the PVZ (Figure 8A), as well as hypertrophied areas of neuroepithelium bulging into the cavity of the brain ventricle, containing numerous individual DC+ cells (Figure 8F). Increased production of DC+ neuroblasts in the post-injury period was found in the PZ, containing large differentiated neurons of the reticular formation (Figure 8B). Such areas were surrounded by a large number of DC+ granules and single DC+ postmitotic neuroblasts, indicating an intensification of migration processes and early neuronal differentiation in the PZ in the post-injury period. In the injury zone, on the contrary, we did not find a large number of DC+ neuroblasts. The predominant type was DC– cells of various size groups, creating dense reactive post-traumatic accumulations of cells (Figure 8C,E). In the PZ adjacent to the area of injury and containing a large number of migrating cells, we revealed a significant increase in the distribution density of DC+ granules, indicating an increase in the synthesis of DC as a protein associated with microtubules and involved in the increasing mobility of migrating cells. Thus, in the injury area, the number of DC+ cells increased slightly; a more significant increase in the number of DC+ cells was found in areas of PVZ remote from the injury, where an increase in the expression of DC in single cells and their groups was also observed. A significant increase in the expression of DC was detected in the PVZ cells surrounding the injury area. The enhanced expression of DC in granules is associated with the increase in cell migration.

## 5. Conclusions

A population of constitutive proliferating progenitor cells located both in the periventricular matrix of the proliferative zone and in the deeper subventricular and parenchymal layers of the brain was revealed in the mesencephalic tegmentum of juvenile chum salmon. As a result of traumatic injury to the tegmentum, proliferation processes in resident neuroepithelial progenitor cells are activated. In the case of damage, along with the reactivation of constitutive aNCS, the direct transformation or dedifferentiation of cells of various origins and degrees of functional maturity with reactivation of genetic proliferation programs is also possible.

Nes+ and Vim+ NPCs and granules located in the periventricular and subventricular matrix zones were identified in the mesencephalic tegmentum of juvenile chum salmon; a high concentration of such cells is localized in the PZ of the tegmentum, which indicates a high level of constructive metabolism and constitutive neurogenesis. After the traumatic process, the expression of Vim and Nes in the aNSCs and NPCs and in the extracellular space significantly increases. As a result of the proliferation of such cells in the post-injury period, local Nes+ and Vim+ NPCs clusters involved in the reparative response are formed. The occurrence of additional nestin and vimentin-positive secondary lesions was revealed in the adjacent SVZ and PZ. Nes+ and Vim+ NPCs in the injury area have a neuroepithelial phenotype. In the lateral part of the tegmentum (*torus semicircularis*), the number of DC+ neuroblasts dominates in comparison with that in the medial zone (DMTN), which is associated with different intensities of primary neuronal differentiation in the sensory and motor regions of the tegmentum, respectively. In the PVZs that are remote from the injury zone, the expression of DC in individual neuroblasts and their groups significantly increases. The number of DC+ cells in the injury area increases slightly. The increased expression of DC in granules is associated with an increase in cell migration.

## Figures and Tables

**Figure 1 brainsci-10-00065-f001:**
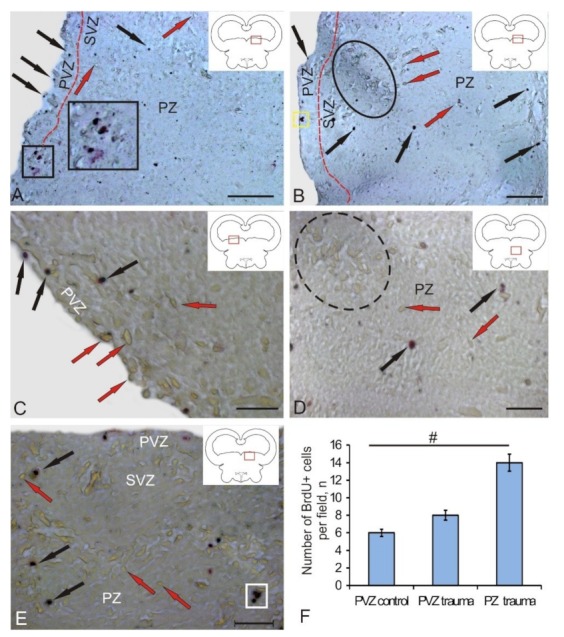
Representative image of BrdU-labeling in the mesencephalic tegmentum of juvenile chum salmon, *Oncorhynchus keta*. (**A**) In the medial tegmentum of intact animals: BrdU+ cells and nuclei (black arrows) and BrdU– cells (red arrows); the border between periventricular zone (PVZ) and subventricular zone (SVZ) is outlined by a red dotted line; accumulation of BrdU+ cells and nuclei at the PVZ/SVZ border is outlined by a black square (inset). (B–E) On day 3 after the traumatic injury to tegmentum: (**B**) in the corresponding zone of the tegmentum after njury; an accumulation of BrdU– cells is shown in the black oval; an accumulation of BrdU+ cells in the PVZ is outlined by yellow dotted line; (**C**) in PVZ; (**D**) in the parenchymal layers (PZ), with an accumulation of BrdU– cells of the reticular formation outlined by black dotted lines; (**E**) in SVZ, with an accumulation of BrdU+ cells shown in white rectangle. Immunoperoxidase BrdU-labeling. (**F**) The quantitative ratio of BrdU+ cells in control animals and after traumatic injury (*n* = 5 in each group; # *p* ≤ 0.05, significant difference vs. control groups); one-way analysis of variance (ANOVA). Scale bar: (A,B) = 100 μm; (C–E) = 50 μm.

**Figure 2 brainsci-10-00065-f002:**
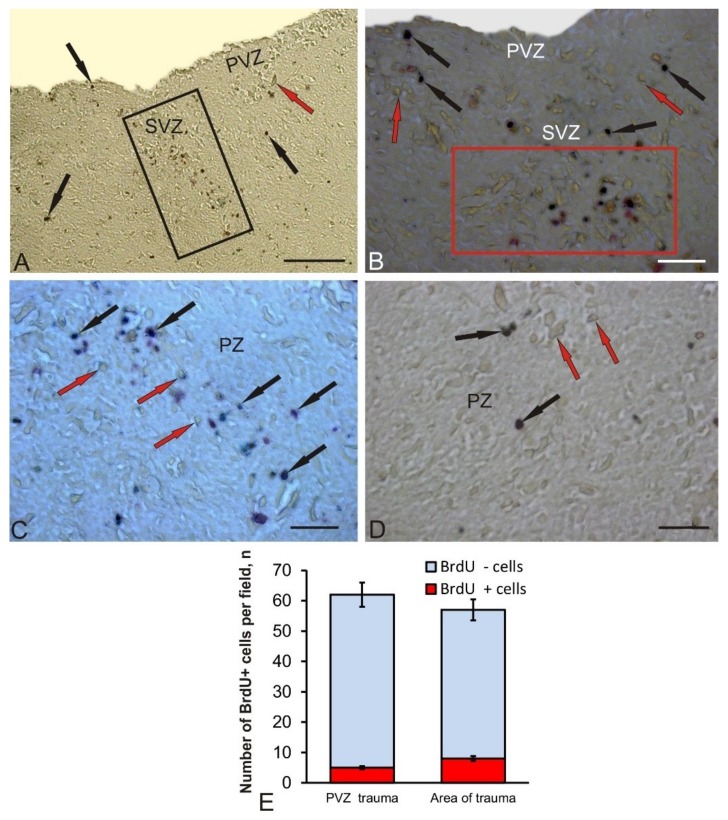
Representative image of BrdU-labeling in the area of injury and adjacent areas of the mesencephalic tegmentum of juvenile chum salmon, *Oncorhynchus keta*, on day 3 post-injury. (**A**) General view of the injury area; a heterogeneous accumulation of BrdU+ cells in the injury zone is outlined by a black rectangle; other designations as in Figure 1. (**B**) Dorsal part of the injury area at high magnification; a heterogeneous accumulation of BrdU+ and BrdU– cells is outlined by a red rectangle. (**C**) The ventral part of the injury area containing heterogeneous BrdU+ cells; (**D**) Near-injury parenchymal zone. Immunoperoxidase BrdU-labeling. (**E**) The ratio of BrdU+/BrdU– cells in the PVZ and the injury area after damage to tegmentum (M ± SD). Scale bar: (A) = 200 μm; (B–D) = 50 μm.

**Figure 3 brainsci-10-00065-f003:**
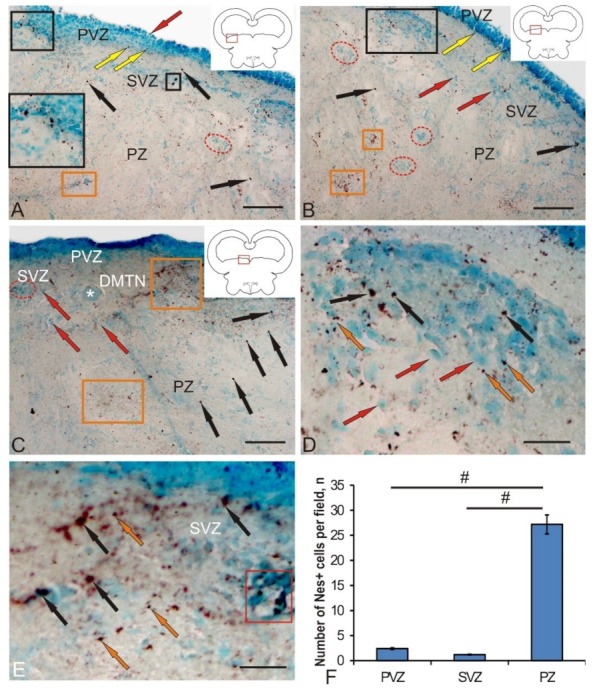
Localization of nestin in the mesencephalic tegmentum of juvenile chum salmon, *Oncorhynchus keta*. (**A**) In the lateral tegmentum: Nes+ cells (black arrows), Nes+ granules (yellow arrows), and Nes– cells (red arrow); an accumulation of Nes+ cells and granules in the PVZ is outlined by black square (inset); constitutive Nes– neurogenic niches (in red dashed ovals) and extracellular deposition of Nes+ granules (in orange rectangles). (**B**) In the dorsal tegmentum; for designations, see Figure 3A. (**C**) In the medial tegmentum: dorso-melial nuclei (DMTN) of the tegmentum; large motoneurons are indicated by a white asterisk. (**D**) An enlarged fragment containing Nes+ cells and granules (inset in Figure 3**B**); Nes+ granules (orange arrows). (**E**) An enlarged fragment of PZ (orange rectangle) in Figure 3**A**. Immunohistochemical (IHC) nestin labeling in combination with methyl green staining. (**F**) The comparative distribution of Nes+ cells in different areas of tegmentum in control groups (*n* = 5 in each group; # are significant intergroup differences); one-way analysis of variance (ANOVA). Scale bar: (A–C) = 100 μm; (D,E) = 50 μm.

**Figure 4 brainsci-10-00065-f004:**
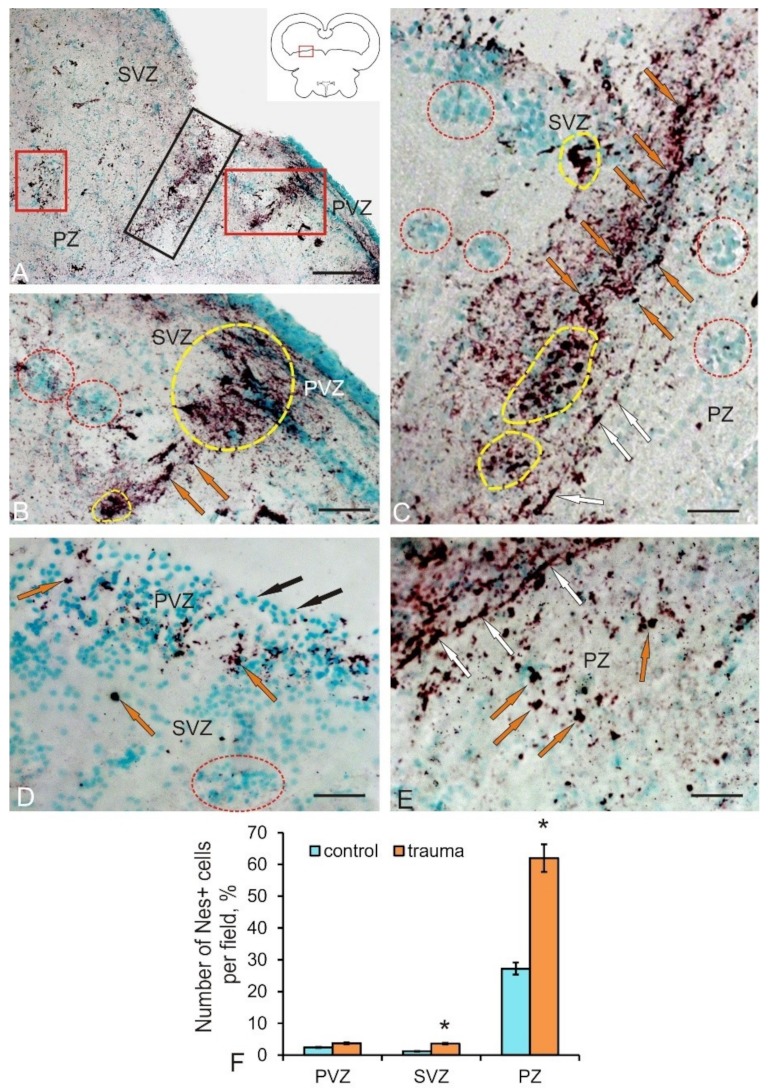
Localization of nestin in the mesencephalic tegmentum of juvenile chum salmon, *Oncorhynchus keta* on day 3 post-injury. (**A**) A general view of the injury zone: black rectangle outlines the central cluster Nes+ NPCs in the injury area; additional Nes+ NPCs clusters are outlined by red rectangles. (**B**) An enlarged fragment of additional Nes+ foci (outlined by yellow dotted lines): reactive neurogenic niches (outlined by red dotted lines) and Nes+ cells (orange arrows). (**C**) An enlarged injury area containing clusters of Nes+ cells (orange arrows) and their clusters inside the traumatic focus Nes+ NPCs located along the Nes– guiding fibers of radial glia (white arrows). (**D**) Nes+ cells of type 4 (orange arrows); reactive groups of negative cells containing Nes+ granules in PVZ and SVZ (orange arrows) and Nes– cells (black arrows) in the near-injury zone. (**E**) Near-injury parenchymal zone containing Nes+ NPCs and Nes– fibers of radial glia (white arrows). IHC nestin labeling in combination with methyl green staining. (**F**) The quantitative ratio of Nes+ cells in the control and after traumatic injury. (*n* = 5 in each group; * *p* ≤ 0.05, significant difference vs. control groups); Student–Newman–Keuls test. Scale bar: (A) = 200 μm; (B–E) = 100 μm.

**Figure 5 brainsci-10-00065-f005:**
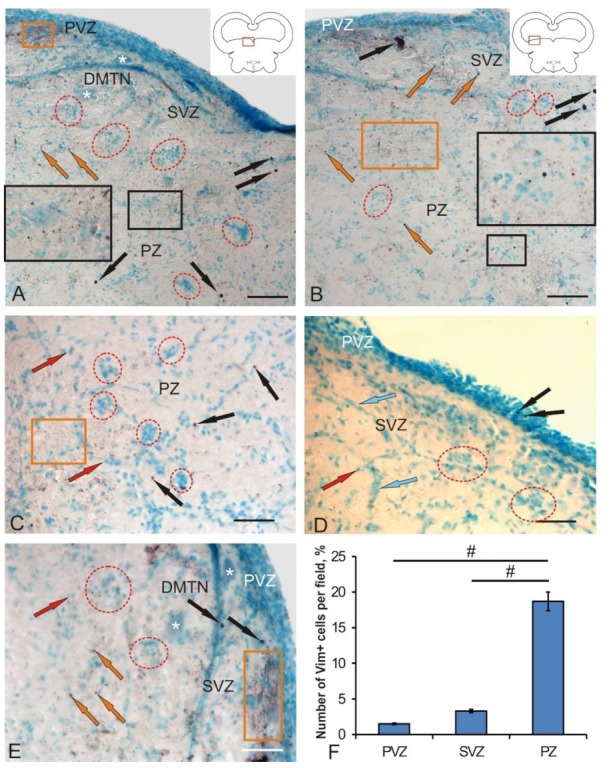
Localization of vimentin in the mesencephalic tegmentum of juvenile chum salmon, *Oncorhynchus keta*. (**A**) In the medial tegmentum: Vim+ cells (black arrows), Vim+ granules (orange arrows), and large neurons in DMTN indicated by white stars, areas of extracellular deposition of vimentin (in an orange rectangle), and constitutive neurogenic niches with Vim– cells (outlined by red dotted lines); Vim+ granules localized both inside cells and in the intercellular space are outlined by black rectangle (inset). (**B**) In the lateral tegmentum (designations as in Figure 5A). (**C**) In the parenchyma of tegmentum: Vim– cells (red arrows). (**D**) In PVZ and SVZ: accumulations of Vim– cells along vessels (blue arrows). (**E**) In DMTN: a dense zone of extracellular deposition of vimentin is outlined by an orange rectangle. IHC vimentin labeling in combination with methyl green staining. (**F**) The ratio of Vim+ cells in various areas of tegmentum in control groups (*n* = 5 in each group; # means significant intergroup differences); one-way analysis of variance (ANOVA). Scale bar: (A,B) = 100 μm; (C–E) = 50 μm.

**Figure 6 brainsci-10-00065-f006:**
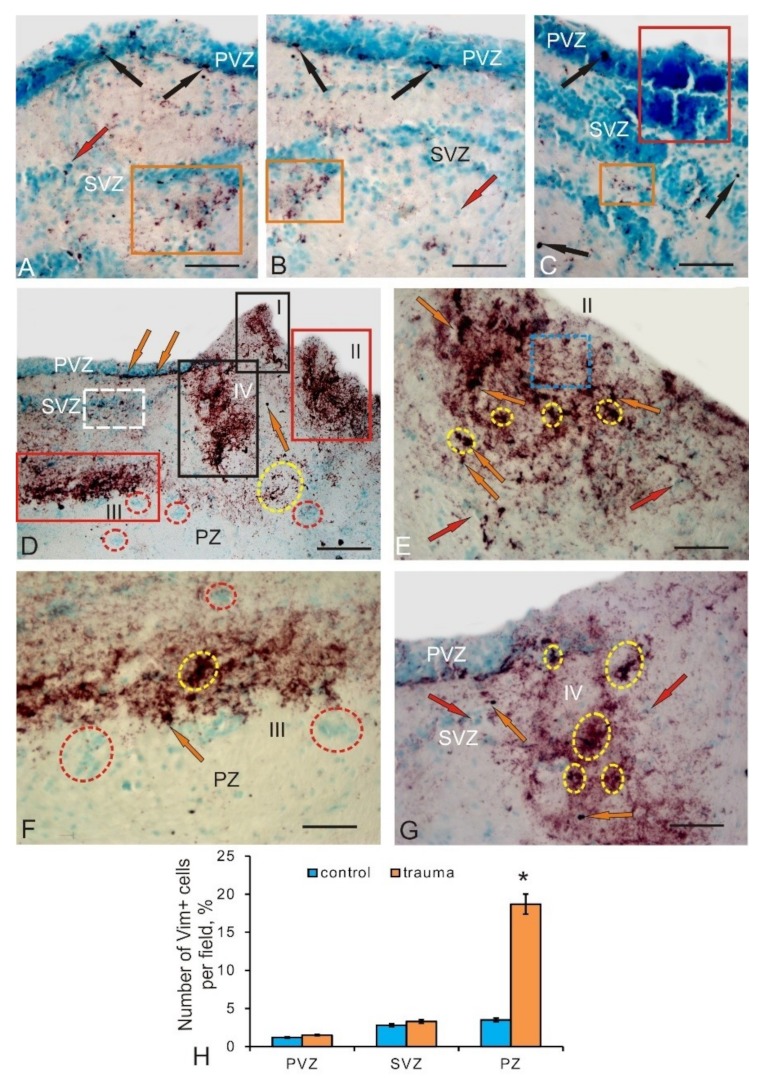
Localization of vimentin in the mesencephalic tegmentum of juvenile chum salmon, *Oncorhynchus keta* on day 3 post-injury. (**A**) In the medial tegmentum: Vim+ NPCs in the basal part of the PVZ (black arrows) and Vim– cells (red arrows); the orange rectangle outlines an accumulation of Vim+ granules in the SVZ. (**B**) In the lateral tegmentum (designation as in Figure 6A). (**C**) In the PVZ adjacent to the area of injury: patterns of increased density of distribution of Vim– cells (outlined by red rectangle), forming hypertrophic accumulation of cells; (**D**) general view of the injured zone and foci (I–IV) of additional vimentin expression: an Vim+ NPCs cluster outlined by a yellow dotted line, reactive neurogenic niches (in red dotted ovals), Vim+ NPCs in the basal part of the PVZ (orange arrows), and a Vim+ NPCs cluster in the SVZ (outlined by a white dotted line). Representative images of foci of additional expression of vimentin. (**E**) In PVZ (II): an extracellular deposition of vimentin is outlined by blue dotted line (other designations as in Figure 6D). (**F**) In PZ (III). (**G**) In SVZ (IV) at a higher magnification. IHC vimentin labeling in combination with methyl green staining. (**H**) The quantitative ratio of Vim+ cells in the control and after traumatic injury. (*n* = 5 in each group; * *p* ≤ 0.05, significant difference vs. control groups); Student–Newman–Keuls test. Scale bar: (A–C,E–G) = 100 μm; (D) = 200 μm.

**Figure 7 brainsci-10-00065-f007:**
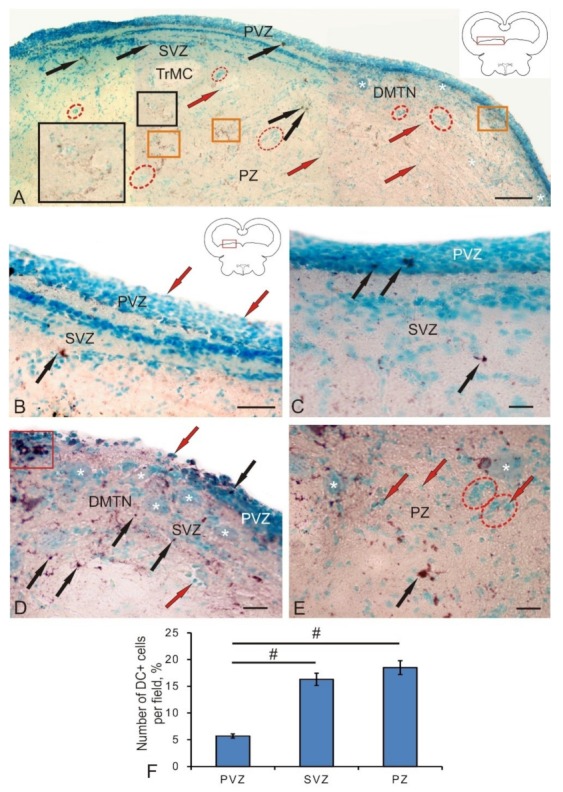
Localization of doublecortin in the mesencephalic tegmentum of juvenile chum salmon, *Oncorhynchus keta*. (**A**) General view of doublecortin (DC) distribution in tegmentum: DC+ cells (black arrows), DC– cells (red arrows), large neurons in DMTN indicated by white stars, mesencephalic-cerebellar tract (TrMC), DC+ granules among DC– cells (outlined by black rectangles) (inset), and extracellular deposition of DC (in orange rectangles); other designations as in Figure 5A. (**B**) In the dorsal tegmentum at a higher magnification. (**C**) In the lateral tegmentum. (**D**) In the medial tegmentum: a cluster of DC+ postmitotic neuroblasts (in a red rectangle). (**E**) In the parenchyma: bodies of large neurons of the reticular formation (indicated by white stars) and constitutive neurogenic niches (red dotted ovals). IHC doublecortin labeling in combination with methyl green staining. (**F**) The ratio of DC+ cells in various areas of tegmentum in control groups (*n* = 5 in each group; # means significant intergroup differences); one-way analysis of variance (ANOVA). Scale bar: (A) = 100 μm; (B) = 50 μm; and (C–E) = 20 μm.

**Figure 8 brainsci-10-00065-f008:**
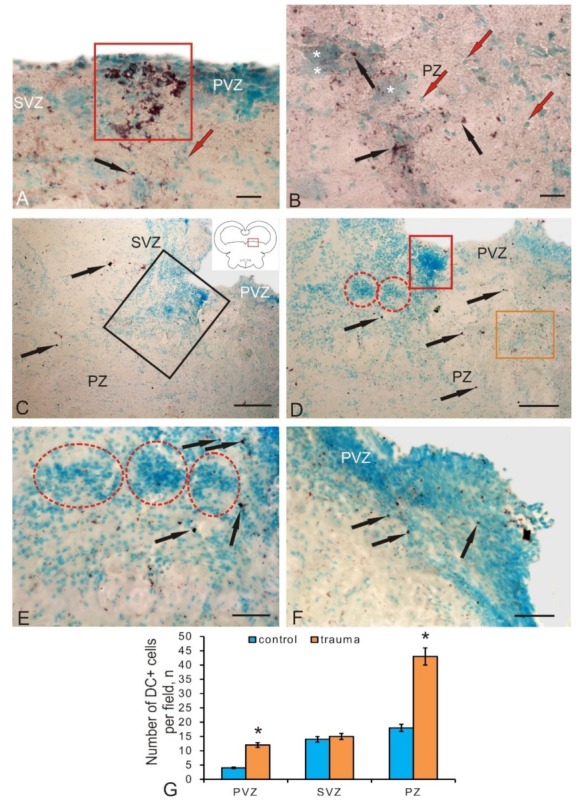
Localization of doublecortin in the mesencephalic tegmentum of juvenile chum salmon, *Onchorhynchus keta* on day 3 days post-injury. (**A**) DC expression in cells (black arrow) and PVZ cell clusters (in the red square); DC– cells (red arrow). (**B**) In the parenchymal near-injury area: large neurons of the reticular formation are indicated by white stars. (**C**) General view of the injury zone (outlined by a black rectangle). (**D**) Dorsal fragment of the injury area: hypertrophy of DC– cells stained with methyl green (in a red rectangle), reactive neurogenic niches (in red dotted ovals), and DC+ neuroblasts in the PZ adjacent to the injury area (in an orange rectangle). (**E**) A ventral fragment of the injury area at high magnification. (**F**) Hypertrophied neuroepithelium in PVZ, containing numerous DC+ neuroblasts. IHC doublecortin labeling in combination with methyl green staining. (**G**) The quantitative ratio of DC+ cells in the control and after traumatic injury (*n* = 5 in each group; * *p* ≤ 0.05, significant difference vs. control groups); Student–Newman–Keuls test. Scale bar: (A,B) = 20 μm; (C) = 200 μm; (D) = 100 μm; (E,F) = 50 μm.

**Figure 9 brainsci-10-00065-f009:**
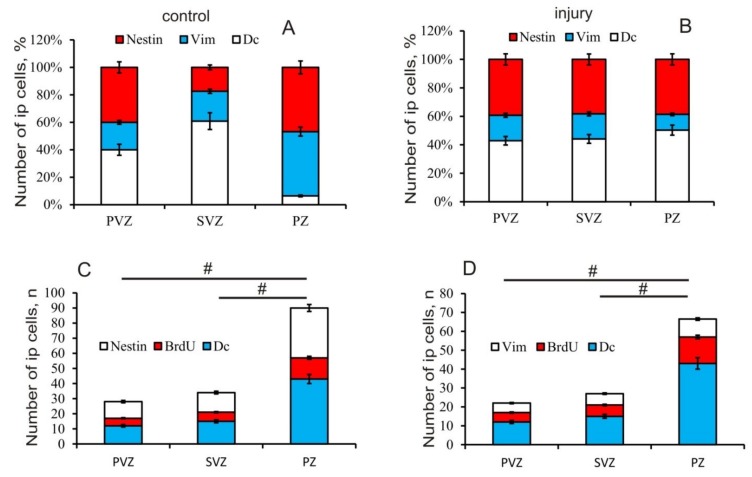
The quantitative ratio of expression of IHC markers of proliferation, NPCs, and neuroblasts, in the intact mesencephalic tegmentum of juvenile chum salmon, *Onchorhynchus keta*, and on day 3 post-injury. (**A**) Expression of nestin, vimentin, and doublecortin in PVZ, SVZ, and PZ in the control (M ± SD). (**B**) After traumatic injury (M ± SD). (**C**) The ratio of nestin+, BrdU+, and DC+ cells in different areas of the chum salmon tegmentum after damage (*n* = 5 in each group; # are significant intergroup differences); one-way analysis of variance (ANOVA). (**D**) The ratio of Vim+, BrdU+, and DC+ cells in different areas of the chum salmon tegmentum after damage (*n* = 5 in each group; # are significant intergroup differences); one-way analysis of variance (ANOVA).

**Table 1 brainsci-10-00065-t001:** Morphometric characteristics of BrdU-labeled cells (M ± SD) in the intact and damaged mesencephalic tegmentum of juvenile chum salmon, *Oncorhynchus keta.*

	PVZ, Intact Animals		Injured Tegmentum	
BrdU-Negative Cells				
Cells/Nuclei	Cell Type	Cell Size, µm *	Cell Type	Cell Size, µm *
Nuclei	1	-	1	-
Undifferentiated	2	5.3 ± 0.5/4.1 ± 0.7	2	-
Oval I	3	6.8 ± 0.6/4.9 ± 0.8	3	7.0 ± 0.7/5.2 ± 1.5
Oval II	4	8.7 ± 0.6/5.3 ± 1.1	4	9.2 ± 1.1/5.9 ± 0.6
Differentiated	5	-	5	14.4 ± 2.3/10.5 ± 1.8
BrdU-positive cells				
Nuclei	1	2.9 ± 0.4/1.7 ± 0.6	1	2.8 ± 0.6/2.0 ± 0.5
Undifferentiated	2	4.5 ± 0.7/3.1 ± 0.9	2	4.3 ± 0.7/2.6 ± 0.7
Oval I	3	7.3 ± 0.5/4.5 ± 0.6	3	6.3 ± 0.2/5.0 ± 0.4
Oval II	4	8.8 ± 0.04/5.5 ± 0.6	4	10.8 ± 0.1/7.9 ± 0.4
Differentiated	5	-	5	13.9 ± 2.6/7.1 ± 1.7

* Large and small diameters of the cell body are shown through a slash.

**Table 2 brainsci-10-00065-t002:** Morphometric characteristics of nestin-labeled cells (M ± SD) in the intact and damaged mesencephalic tegmentum of juvenile chum salmon, *Oncorhynchus keta.*

	Intact Animals		Injured Tegmentum	
Nes-Negative Cells				
Cells/Granules	Cell Type	Cell Size *, µm	Cell Type	Cell Size *, µm
Granules	1	-	1	-
Undifferentiated	2	5.3 ± 0.5/3.4 ± 0.6	2	5.3 ± 0.5/3.5 ± 0.7
Oval I	3	6.9 ± 0.6/3.6 ± 0.9	3	6.7 ± 0.5/4.4 ± 0.8
Oval II	4	10.2 ± 1.8/4.4 ± 1.3	4	8.3 ± 0.4/4.2 ± 1.3
Differentiated	5	21.7 ± 10.9/12 ± 11.5	5	-
Nes-positive cells				
Granules	1	2.5 ± 0.6/1.6 ± 0.5	1	2.9 ± 0.3/2.1 ± 0.4
Undifferentiated	2	4.2 ± 0.5/2.7 ± 0.7	2	4.5 ± 0.6/3.1 ± 0.9
Oval I	3	-	3	6.9 ± 0.4/4.4 ± 1.0
Oval II	4	-	4	9.8 ± 1.3/6.7 ± 0.6
Differentiated	5	-	5	-

* Large and small diameters of the cell body are shown through a slash.

**Table 3 brainsci-10-00065-t003:** Morphometric characteristics of vimentin-labeled cells (M ± SD) in the intact and damaged mesencephalic tegmentum of juvenile chum salmon, *Oncorhynchus keta.*

	Intact Animals		Injured Tegmentum	
Vim-Negative Cells				
Cells/Granules	Cell Type	Cell Size *, µm	Cell Type	Cell Size *, µm
Granules	1	-	1	-
Undifferentiated	2	5.4 ± 0.4/3.8 ± 0.7	2	5.3 ± 0.5/3.5 ± 0.6
Oval I	3	6.9 ± 0.5/4.1 ± 0.8	3	6.9 ± 0.5/4.1 ± 0.7
Oval II	4	9.4 ± 1.2/4.3 ± 1.1	4	9.8 ± 1.4/4.5 ± 1.2
Differentiated	5	20.4 ± 3.5/4.2 ± 0.8	5	-
Vim-positive cells				
Granules	1	2.6 ± 0.6/1.6 ± 0.4	1	2.7 ± 0.6/1.7 ± 0.6
Undifferentiated	2	4.2 ± 0.5/2.8 ± 0.6	2	4.8 ± 0.7/3.1 ± 0.7
Oval I	3	6.8 ± 0.5/4.8 ± 1.1	3	6.8 ± 0.6/4.0 ± 0.9
Oval II	4	10.5 ± 2.5/7.6 ± 2.5	4	-
Differentiated	5	-	5	-

* Large and small diameters of the cell body are shown through a slash.

**Table 4 brainsci-10-00065-t004:** Morphometric characteristics doublecortin-labeled cells (M ± SD) in the intact and damaged mesencephalic tegmentum of juvenile chum salmon, *Oncorhynchus keta.*

	Intact Animals		Injured Tegmentum	
DC-Negative Cells				
Cells/Granules	Cell Type	Cell Size *, µm	Cell Type	Cell Size *, µm
Granules	1		1	
Undifferentiated	2	-	2	5.5 ± 0.4/4.3 ± 0.8
Oval I	3	7.3 ± 0.5/4.5 ± 0.7	3	7.0 ± 0.6/4.8 ± 0.7
Oval II	4	9.8 ± 1.3/5.1 ± 0.9	4	9.4 ± 1.3/5.5 ± 1.0
Differentiated	5	14.1 ± 1.6/6.4 ± 1.2	5	
DC- positive cells				
Granules	1	3.5 ± 0.4/2.6 ± 0.5	1	2.9 ± 0.4/2.1 ± 0.3
Undifferentiated	2	4.6 ± 0.6/2.9 ± 0.6	2	5.0 ± 0.7/3.4 ± 0.8
Oval I	3	7.1 ± 0.5/3.9 ± 1.2	3	6.9 ± 0.5/4.3 ± 0.9
Oval II	4	8.6 ± 0.5/5.1 ± 1.2	4	9.9 ± 1.3/5.6 ± 1.2
Differentiated	5	-	5	-

* Large and small diameters of the cell body are shown through a slash.

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
