# Peer review of "Neural Stem Cells/Neuronal Precursor Cells and Postmitotic Neuroblasts in Constitutive Neurogenesis and After ,Traumatic Injury to the Mesencephalic Tegmentum of Juvenile Chum Salmon, Oncorhynchus keta"

_brainsci, 2020, doi:10.3390/brainsci10020065_

Round 1
Reviewer 1 Report
Authors characterize the progenitor and neuroblast populations in the mesencephalic tegmentum area and their response to traumatic injury in the juvenile chum salmon. I have some minor comments
1-What does DC stand for in the abstract? Please use the open form first before the abbreviation.
2-Please correct the spelling travma = trauma.
Overall in the images (particularly figure 1 and 2), it is very hard to understand how the authors quantified the Brdu- nuclei. Demonstration of what a nuclei (or cell) is with a clear counterstain for a nuclei acid marker or other markers that will allow identification of cells would be very helpful.
Please show the non injured control at matched levels/location in Figure 2, 4, 6. I understand these are shown in the other figures but it is hard to interpret the data and see the injury induced changes without seeing them side by side.
Author Response
Response to Reviewer 1 Comments
Authors characterize the progenitor and neuroblast populations in the mesencephalic tegmentum area and their response to traumatic injury in the juvenile chum salmon. I have some minor comments
Point 1: 1-What does DC stand for in the abstract? Please use the open form first before the abbreviation.
Response 1: Corrections in accordance with the remark made in the abstract
Point 2: 2-Please correct the spelling travma = trauma.
Response 2: Corrections in accordance with the remark made in the all Figures
Point 3: Overall in the images (particularly figure 1 and 2), it is very hard to understand how the authors quantified the Brdu- nuclei. Demonstration of what a nuclei (or cell) is with a clear counterstain for a nuclei acid marker or other markers that will allow identification of cells would be very helpful.
Response 3: A quantitative assessment of BrdU of labeled nuclei was carried out by counting labeled elements (cells and nuclei) on sections of the brain in intact animals and after traumatic damage to tegmentum, with an increase in the lens 20X and the eyepiece 10X. The field of view was taken as the profile field (per field), on which the number (n) of labeled cells and nuclei was counted (see classification Table 1). In the control group and after injury, there were 5 animals. Brain sections during IHC labeling of BrdU were not additionally stained with nuclear dye, however immunonegative cells (Fig. 1C, red arrows) and their clusters (Fig. 1D are outlined by a dotted oval) are quite clearly visible at high magnification (x40).
Point 4: Please show the non injured control at matched levels/location in Figure 2, 4, 6. I understand these are shown in the other figures but it is hard to interpret the data and see the injury induced changes without seeing them side by side.
Response 4: In the present work, illustrative material is grouped so that in Fig. 1,3,5,7 are illustrations showing representative images of the distribution of the IHC marking of Brdu, Nes, Wim and Dk in intact (control) animals. In fig. 2,4,6 and 8 are images of patterns of distribution and quantification of these markers 3 days after the injury of tegumentum. It seems to us logical to group the material in such a way that the data on the distribution of markers in the constitutive process alternate with the data in the reparative response.

Reviewer 2 Report
The article “Neural Stem Cells/Neuronal Precursor Cells and Postmitotic Neuroblasts in Constitutive Neurogenesis and After Traumatic Injury to the Mesencephalic Tegmentum of Juvenile Chum Salmon, Oncorhynchus keta.” By Pushchina et al presented a study to wound the Mesencephalic Tegmentum of Juvenile Chum Salmon and characterized the cell proliferation and cell phenotype in neurogenesis. The study demonstrates some interesting observation. The concerns are listed.
The co-labeling of the BrdU and other neural markers should be performed to determine the cell phenotype of proliferated cells. It is not clear how many sections were made in the injury area. The authors should describe the criteria for the selection of the section and the cell number counting. Please describe how the differentiated and undifferentiated cells are determined in table 1 and 2. The figures 5F and 6 H are confusing. Is the figure 5F calculation of the control or trauma group? It should be clarified.
Author Response
Response to Reviewer 2 Comments
The article “Neural Stem Cells/Neuronal Precursor Cells and Postmitotic Neuroblasts in Constitutive Neurogenesis and After Traumatic Injury to the Mesencephalic Tegmentum of Juvenile Chum Salmon, Oncorhynchus keta.” By Pushchina et al presented a study to wound the Mesencephalic Tegmentum of Juvenile Chum Salmon and characterized the cell proliferation and cell phenotype in neurogenesis. The study demonstrates some interesting observation. The concerns are listed.
Point 1: The co-labeling of the BrdU and other neural markers should be performed to determine the cell phenotype of proliferated cells. It is not clear how many sections were made in the injury area. The authors should describe the criteria for the selection of the section and the cell number counting. Please describe how the differentiated and undifferentiated cells are determined in table 1 and 2. The figures 5F and 6 H are confusing. Is the figure 5F calculation of the control or trauma group? It should be clarified.
Response 1: We agree with the remark regarding the determination of the phenotype of proliferating cells and a corresponding correction was made in the formulation of the purpose of the work. In the area of brain injury, at least 10 sections from 5 animals for each marker were made and analyzed. The separation of cell types was carried out in accordance with morphotopographic criteria and specific labeling (Brdu and Nest), in accordance with the previously developed classification (16), which was based on the measurement of the large and small diameter of the soma of the cell, which allows to determinate small undifferentiated cells, several groups of oval cells and large differentiated cells. Additional staining of cells with methyl green made it possible to identify and measure immuno-negative cells, the sizes of which are shown in tables 1 and 2. The corresponding additions were made in section 2.5. Microscopy, section and 2. Material and methods. The number of cells was calculated on the profile field with an increase in the lens 20x, eyepiece 10x. Figures 5F and 6 H have been updated accordingly.